# Lightweight Super-Resolution Techniques in Medical Imaging: Bridging Quality and Computational Efficiency

**DOI:** 10.3390/bioengineering11121179

**Published:** 2024-11-21

**Authors:** Akmalbek Abdusalomov, Sanjar Mirzakhalilov, Zaripova Dilnoza, Kudratjon Zohirov, Rashid Nasimov, Sabina Umirzakova, Young-Im Cho

**Affiliations:** 1Department of Computer Engineering, Gachon University Sujeong-Gu, Seongnam-Si 13120, Gyeonggi-Do, Republic of Korea; bobomirzaevich@gmail.com; 2Department of Computer Systems/Information and Educational Technologies, Tashkent University of Information Technologies Named After Muhammad Al-Khwarizmi, Tashkent 100200, Uzbekistan; mirzaxalilov86@tuit.uz (S.M.); zaripovada85@gmail.com (Z.D.); 3Department of Computer Systems, Karshi Branch of the Tashkent University of Information Technologies Named After Muhammad al-Khwarizmi, Tashkent 100200, Uzbekistan; qzohirov@gmail.com; 4Department of Artificial Intelligence, Tashkent State University of Economics, Tashkent 100066, Uzbekistan; rashid.nasimov@tsue.uz

**Keywords:** medical imaging, super-resolution, lightweight model, residual learning

## Abstract

Medical imaging plays an essential role in modern healthcare, providing non-invasive tools for diagnosing and monitoring various medical conditions. However, the resolution limitations of imaging hardware often result in suboptimal images, which can hinder the precision of clinical decision-making. Single image super-resolution (SISR) techniques offer a solution by reconstructing high-resolution (HR) images from low-resolution (LR) counterparts, enhancing the visual quality of medical images. In this paper, we propose an enhanced Residual Feature Learning Network (RFLN) tailored specifically for medical imaging. Our contributions include replacing the residual local feature blocks with standard residual blocks, increasing the model depth for improved feature extraction, and incorporating enhanced spatial attention (ESA) mechanisms to refine the feature selection. Extensive experiments on medical imaging datasets demonstrate that the proposed model achieves superior performance in terms of both quantitative metrics, such as PSNR and SSIM, and qualitative visual quality compared to existing state-of-the-art models. The enhanced RFLN not only effectively mitigates noise but also preserves critical anatomical details, making it a promising solution for high-precision medical imaging applications.

## 1. Introduction

Medical imaging plays a crucial role in modern healthcare, providing non-invasive methods for diagnosing, monitoring, and treating various medical conditions. With the ever-increasing demand for higher precision in medical image interpretation, enhancing the resolution of medical images has become a fundamental task [1]. High-resolution medical images provide better visualization of intricate anatomical details, aiding healthcare professionals in making accurate diagnoses and planning effective treatments [2]. However, the limitations of imaging hardware often lead to sub-optimal image resolution, making the development of advanced super-resolution techniques imperative.

Single image super-resolution (SISR) aims to reconstruct a high-resolution (HR) image from a given low-resolution (LR) image, recovering fine details that are essential for accurate analysis [3]. In recent years, deep learning-based approaches have demonstrated significant improvements in SISR tasks, surpassing traditional methods in both visual quality and computational efficiency [4]. This paper focuses on enhancing the Residual Feature Learning Network (RFLN) [5] for medical imaging by proposing modifications tailored specifically for the unique challenges posed by medical datasets [6]. The proposed model, an enhanced version of the RFLN, replaces the original residual local feature block with a standard residual block and introduces deeper layers for improved feature extraction, which is particularly suited for medical images, that often have only a single input channel. Our modifications aim to retain essential features while reducing irrelevant noise, thereby providing high-fidelity reconstructions, which are critical for medical applications. This paper details the architectural changes, the training methodology, and a comprehensive comparison with other state-of-the-art super-resolution models, demonstrating the superiority of the proposed model, both in terms of performance metrics and the visual results of medical imaging tasks.

The main contributions of this work are summarized as follows:We propose an improved version of the RFLN architecture tailored specifically for medical imaging. The modifications include replacing the residual local feature blocks with standard residual blocks and increasing the model depth to improve feature extraction and resolution quality.To further improve the feature refinement process, we integrate an enhanced spatial attention mechanism into the model. This helps focus on the most relevant areas of the input image, enhancing the overall quality of the super-resolved output, which is crucial for medical image interpretation.We conduct extensive experiments using specialized medical imaging datasets, demonstrating the efficacy of the proposed model in terms of both quantitative metrics and qualitative results. The proposed model outperforms existing state-of-the-art models, showcasing its potential for real-world medical applications.We provide a detailed analysis of the proposed model’s performance compared to other leading SISR models. Our results highlight the advantages of our modifications in handling the unique challenges presented by medical images, such as the preservation of subtle anatomical details and noise reduction.

These contributions collectively advance the field of super-resolution for medical imaging, providing a robust framework that can significantly enhance the quality of medical images, ultimately aiding healthcare professionals in delivering accurate and effective diagnoses. In the following sections, we outline the structure of this paper: Section 2 reviews related works in the field of single image super-resolution (SISR). Section 3 presents the methodology, detailing the proposed enhancements to the RFLN. Section 4 describes the experiments and results, while Section 5 discusses the findings and implications. Finally, Section 6 concludes the paper by summarizing the main contributions and future directions.

## 2. Related Works

The field of SISR has seen considerable advancements in recent years, primarily due to the advent of deep learning-based methods [7]. Early approaches, such as bicubic interpolation [8] and sparse representation-based methods [9], laid the groundwork for SISR but were limited in their ability to capture complex textures and fine details. Traditional machine learning techniques, such as Sparse Coding-based Super-Resolution (SC-SR) [10] and Neighbor Embedding-based Super-Resolution (NE-SR) [11], provided some improvements over basic interpolation techniques but still struggled with high-frequency detail restoration. The introduction of convolutional neural networks (CNNs) marked a turning point for SISR. Dong et al. introduced the Super-Resolution Convolutional Neural Network (SRCNN) [3], which was among the first to demonstrate the potential of deep learning for super-resolution tasks. SRCNN utilized a straightforward architecture, achieving notable improvements in visual quality over traditional methods. Subsequent models, such as Very Deep Super-Resolution (VDSR) [12] and the Deep Recursive Convolutional Network (DRCN) [13], leveraged deeper architectures and recursive structures to further improve super-resolution performance. Generative adversarial networks (GANs) have also been employed to enhance the perceptual quality of super-resolved images [14]. The SRGAN [15] model was a pioneering effort that introduced adversarial loss to encourage the generation of high-frequency details [16], producing images that were perceptually closer to the ground truth. However, GAN-based models often suffer from instability during training and may introduce artifacts that compromise the accuracy of medical images, where precise detail is crucial.

More recent works have focused on residual learning and attention mechanisms [17] to address the challenges of high-fidelity image reconstruction. The authors of [18] introduced the Enhanced Deep Residual Network for Single Image Super-Resolution (EDSR), which discarded unnecessary layers and batch normalization to achieve better performance. The authors of [19] proposed the Residual Channel Attention Network (RCAN) [20], which leveraged channel-wise attention to adaptively refine features, leading to significant improvements in image quality [21]. These models have demonstrated the importance of focusing on key features while suppressing irrelevant information, an approach that aligns closely with the goals of our proposed method. The RFLN, which serves as the baseline for our work, utilizes residual local feature blocks to extract both local and global features [22]. The RFLN has shown promise in maintaining a stable gradient flow and capturing intricate details, making it effective for general SISR tasks [23]. However, medical imaging presents unique challenges, such as the need for high precision in areas with subtle anatomical details, which necessitates specialized modifications [24]. 

In the domain of medical imaging, several models have been proposed to enhance the quality of medical images specifically. The authors of [25] proposed a deep learning framework for MRI super-resolution [26], demonstrating the effectiveness of tailored loss functions for medical data. The authors of [27] introduced a semi-supervised framework, Mine your own Anatomy (MONA), which strategically utilizes dataset characteristics for improved segmentation. The authors of [28] presented a fuzzy neural block that converts images into a fuzzy domain, processes pixel uncertainty with fuzzy rules, and fuses these results with standard convolutional outputs. The authors of [29] proposed a novel Multimodal Multi-Head Convolutional Attention (MMHCA) module to enhance super-resolution for these scans. The module jointly applies spatial-channel attention via convolutions on concatenated input tensors, where the kernel size controls the spatial attention reduction and the number of filters manages the channel attention reduction. The authors of [30] proposed a novel UMIE approach that encodes HQ features directly into the enhancement process using a variation normalization module. This joint modeling of LQ and HQ domains ensures better guidance. The network is trained adversarial with a discriminator to ensure the output belongs to the HQ domain. The work outlined in [31] enhances traditional SR methods by incorporating a channel attention block specifically designed for high-frequency features, which are critical for detailed medical diagnostics. DRFDCAN utilizes a residual-within-residual architecture to improve inference speed and reduce memory usage without compromising image quality. The problem addressed in this study is the ability to enhance low-resolution medical images to high-resolution quality while maintaining computational efficiency. The limitations of existing medical imaging hardware often lead to images that lack sufficient resolution for precise clinical diagnosis. While recent advancements in SISR using deep learning have shown promise, many state-of-the-art models are computationally intensive and not suitable for practical deployment in medical environments. Therefore, the challenge lies in developing a lightweight, efficient SISR model that can effectively enhance image resolution without compromising quality or requiring extensive computational resources. This study aims to address these challenges by proposing an enhanced RFLN specifically tailored for medical imaging.

Previous methods in medical image enhancement face several limitations that are addressed by our model [32]. Many, like MRI super-resolution and MONA, struggle with generalization across different modalities and rely heavily on specific datasets [33]. Our model, with its deeper architecture and standard residual blocks, adapts better across various medical images. Additionally, noise mitigation in earlier approaches, such as fuzzy neural blocks, is less effective, while our model targets noise reduction without sacrificing important features. Computational complexity is another drawback, particularly in models like MMHCA and DRFDCAN, which are resource-intensive [34]. Our model streamlines this by using enhanced spatial attention for efficient processing. Moreover, methods like DRFDCAN can overemphasize details at the cost of larger structural integrity, which our approach balances. Finally, unlike adversarial-based techniques like UMIE, which can be unstable, our model offers stable, consistent results, without introducing artifacts. Our model provides a more efficient, adaptable, and reliable solution for medical imaging. 

Our proposed model builds on these advancements by enhancing the RFLN architecture to better suit medical imaging applications. By replacing the residual local feature blocks with standard residual blocks and increasing the model depth, our approach aims to retain critical features while effectively mitigating noise. Additionally, the incorporation of enhanced spatial attention (ESA) mechanisms ensures that the model can focus on the most relevant features, thereby improving both the visual quality and diagnostic utility of the reconstructed images. 

## 3. Methodology

In this section, we present the enhanced RFLN model tailored for single image super-resolution. In our work, we specifically adapted the model for medical imaging by replacing its residual local feature block with a standard residual block. Section 3.1 provides a comprehensive overview of the baseline model, while Section 3.2 offers a detailed structural analysis of the proposed modifications and loss functions. 

### 3.1. Residual Local Feature Network

The RFLN is a deep learning model primarily designed to enhance single image super-resolution tasks (Figure 1). It leverages the concept of residual learning, which facilitates the ability to focus on learning the residual (or difference) between the LR input and the HR output, rather than attempting to directly reconstruct the high-resolution image. This technique has proven effective in mitigating vanishing gradient issues, allowing for more efficient and accurate deep network training. At the core of the RFLN architecture are residual local feature blocks, which are designed to extract and preserve both local and global features from the input image. These blocks work in synergy to capture intricate details across various scales of the image, ensuring that the reconstructed high-resolution image maintains sharpness and fine texture details. The architecture also incorporates several layers of convolutional operations, each followed by non-linear activation functions, which together contribute to the progressive refinement of the image resolution. 

The residual local feature blocks within the network are particularly adept at retaining essential image features while suppressing irrelevant noise, which is critical for high-quality image super-resolution. In essence, RFLN focuses on learning local dependencies within the image, while the residual connections across layers ensure that the network can maintain stable gradient flow during training, preventing performance degradation in deeper layers. Furthermore, the integration of upsampling techniques toward the later stages of the network enables the final generation of the high-resolution output. These methods, often involving pixel shuffling or deconvolution, ensure that the output resolution is increased efficiently without introducing significant artifacts. By leveraging these techniques, RFLN can achieve precise and visually appealing super-resolution, making it particularly effective for applications that require high-fidelity image reconstruction.

### 3.2. The Proposed Model

where each block consists of three convolutional layers with 3 × 3 kernel sizes, each paired with ReLU activation functions to introduce non-linearity to the feature maps (Figure 2). Following this, a concatenation layer combines the output of the block with the input feature map.

These steps are followed by the addition of another convolutional layer and an ESA block, which contribute to the enhancement of inference time. The incorporation of the ESA block in the proposed architecture is essential for effectively refining feature selection and improving the quality of the reconstructed images. In medical imaging, preserving subtle anatomical details is critical for accurate diagnosis, and the ESA block helps the model focus on these crucial regions of the input images. The ESA block operates by selectively emphasizing significant spatial areas, which enhances the model’s ability to retain important features while mitigating irrelevant noise. By capturing both local and global dependencies, the ESA block allows the model to distinguish between essential anatomical information and less relevant background features. This targeted attention is particularly beneficial for medical images, where precise feature extraction can greatly influence clinical outcomes. The use of the ESA block ultimately results in higher-quality super-resolved images with superior visual fidelity, making it an integral component of the model architecture for medical applications. 

In our modified ResBlocks, we restructure the entire model, slightly increasing its depth to capture more comprehensive information, as we are working with medical images that typically have only a single input channel. This adjustment allows for more effective feature extraction, ensuring that critical details are retained, which is essential for the precision required in medical image processing. The input image, ${X_{input}\in R}^{WxHxC}$, goes as the first layer into ResBlock1, as shown in Equation (1):(1)$$F_{layer1}=F_{1x1}(X_{input})$$

$F_{layer1}$ constitutes the initial layer, which comprises a single convolutional layer with a 3 × 3 kernel size, designed to extract low-level features:(2)$$F_{layer2}=max(0,x\cdot (BatchNorm(F_{3x3}(F_{layer1}))))$$

Equation (2), where $F_{layer2}$ denotes the second layer, incorporates a 3 × 3 convolution layer that enhances feature mapping. This is succeeded by batch normalization, which stabilizes the learning process by normalizing the input layer by re-centering and re-scaling. Following batch normalization, the ReLU activation function is applied to introduce non-linearity, facilitating the ability of the model to learn complex patterns in the data:(3)$$F_{layer3}=max(0,x\cdot ((F_{3x3}(F_{layer1})))$$
(4)$$F_{layer4}=F_{1x1}(F_{concat}(F_{layer2},F_{layer3}))$$

Equation (3) delineates the layers equipped with a ReLU activation function and a 3 × 3 convolution layer, tailored for extracting coarser features. Concurrently, Equation (4) illustrates the process of element-wise concatenation, coupled with a layer dedicated to the extraction of low-level features:(5)$$F_{layer5}=max(0,x\cdot (BatchNorm(F_{3x3}(F_{layer4}))))$$
(6)$$F_{layer6}=max(0,x\cdot ((F_{3x3}(F_{layer4})))$$

Equations (5) and (6) replicate the same blocks as those in Equations (2) and (3). In these instances, the input feature map, following concatenation, restores some information and possesses more complex features for subsequent feature extraction layers. Additionally, element-wise concatenation aids the feature map in preserving essential information, preventing the loss of crucial details:(7)$$F_{layer7}=F_{concat}(F_{layer6},F_{layer5})$$
(8)$$F_{layer8}=F_{1x1}(MaxPooling(F_{layer7}))$$
(9)$$F_{layer9}=F_{ESA}(F_{layer8})$$

In Equation (7), we employ the second concatenation layer, followed by the application of a pooling layer. This step ensures that the model captures the most essential or representative features from the input feature maps. Specifically, we utilize max pooling, which selects the maximum value from a group of pixels within a feature map. This approach is effective because the maximum value typically represents a distinct feature or shape characteristic, aiding in the robust recognition and representation of important spatial hierarchies in the data. In the refinement stages, we employ the Mean Squared Error (MSE) loss. This metric is pivotal in diminishing the squared discrepancies between the predicted and actual pixel values, which is integral for augmenting the precision of super-resolution outcomes. The MSE loss quantifies the average squared variances between the true and forecasted values. It is conventionally articulated as follows:(10)$$L_{MSE}=\frac{1}{N}\sum_{i}{\left(y_{i}-{y^{\prime }}_{i}\right)}^{2}$$

This expression underscores the aim to minimize the mean of the squared errors, thereby enhancing the fidelity and quality of the super-resolved images. In Equation (10), $y_{i}$ represents the ground truth high-resolution images, while ${y^{\prime }}_{i}$ denotes the predicted high-resolution images. These variables are crucial for assessing the performance of super-resolution models, focusing on reducing the discrepancies between the actual and computed outputs to enhance image quality.

## 4. Experiments and Results

### 4.1. The Dataset

In our research, we harness specialized datasets that are designed specifically for super-resolution tasks, such as Figshare and the Kidney Stone collections. These datasets are expertly structured to facilitate the training and evaluation of sophisticated super-resolution models. As integral components of our study, they include a comprehensive array of images meticulously prepared and annotated to support the development of cutting-edge image enhancement technologies. These datasets are instrumental in advancing techniques that precisely enhance image resolution, catering to the unique demands of super-resolution applications. Each dataset includes a diverse set of images from various scenarios, enriched with detailed annotations that delineate critical areas within the images, thereby providing high-quality data crucial for refining model accuracy and performance. The characteristics of the datasets used for testing, namely Figshare and Kidney Stone, are presented below. These datasets were carefully selected to provide a diverse range of medical images, ensuring that the proposed model could be effectively evaluated across different medical imaging modalities (Table 1).

Table 1 provides a clear overview of the datasets used, detailing the number of images, their resolution, the imaging modality, and a brief description of the dataset content. Including this information allows for better reproducibility of this study and transparency regarding the data used.

### 4.2. Data Preprocessing

The data preprocessing phase for the Figshare and Kidney Stone datasets, which are tailored for medical super-resolution tasks, involves a pipeline rigorously designed to enhance the robustness and accuracy of the proposed model. Initially, all images are resized to ensure uniformity, meeting the specific input requirements of the proposed model and maintaining consistency across the dataset. Considering the variability in medical imaging, such as differing modalities and scan qualities, we introduce a series of augmentations. These adjustments include random rotations, flips, and variations in brightness and contrast, which help to mimic the diverse conditions found in real-world medical settings. Additionally, to replicate common imaging challenges like noise and slight blurring, which may occur due to machine imperfections or patient movement, Gaussian noise and blurring are applied. Through this preparation process we aim to acclimate the model to potential real-world imperfections it might encounter. 

Normalization of each image follows, standardizing pixel values to align with the neural expectations of the proposed model, thus promoting stable and efficient learning. If the datasets include specific annotations, such as regions of interest around kidney stones, these are meticulously adjusted to maintain accuracy after image transformations. Lastly, to ensure comprehensive training and prevent model bias, we balance the dataset by employing techniques like oversampling or undersampling to represent various medical conditions adequately (Figure 3).

### 4.3. Metrics

In this paper, the primary metrics we use to evaluate the model’s performance are the Peak Signal-to-Noise Ratio (PSNR) and the Structural Similarity Index Measure (SSIM). These metrics are crucial for assessing the quality of the images that have been enhanced by the super-resolution process. PSNR is used to measure the ratio between the maximum possible power of a signal and the power of any corrupting noise that affects the fidelity of its representation, which, in image processing, translates to how much detail can be perceived in the super-resolved image:(11)$$PSNR=20\cdot {log}_{10}(\frac{{MAX}_{I}}{\sqrt{MSE}})$$
where ${MAX}_{I}$ is the maximum possible pixel value of the image (e.g., 255 for 8-bit images) and MSE is the mean squared error between the original and the reconstructed images. SSIM, on the other hand, evaluates the visual impact of three characteristics of an image (luminance, contrast, and structure), thereby providing a more comprehensive measure of image quality and perceived changes in structural information. These metrics are instrumental in demonstrating the effectiveness of the proposed model in improving the resolution of medical images while maintaining a balance between computational efficiency and enhancement quality:(12)$$SSIM(x,y)=\frac{\left(2\mu_{x}\mu_{y}+c_{1})(2\sigma_{xy}+c_{2}\right)}{\left(\mu_{x}^{2}+\mu_{y}^{2}+c_{1})(\sigma_{x}^{2}+\sigma_{y}^{2}+c_{2}\right)}$$
where $\mu_{x}\;$and $\mu_{y}$ are the averages of *x* and *y*, respectively, $\sigma_{x}$, and $\sigma_{y}$ are the variance of *x* and *y*, respectively, $\sigma_{xy}$ is the covariance of *x* and *y*, and $c_{1}$ and $c_{2}$ are two variables to stabilize division with a weak denominator.

### 4.4. Experimental Results

Figure 4 and Figure 5 illustrate a meticulous evaluation of super-resolution techniques applied to a medical MRI image, focusing on a patch region within the brain. These figures display a sequence of images showing the original scan and subsequent enhancements using various super-resolution models, including SRCNN, SRGAN, VDSR [12], a baseline model, and the proposed model.

The sequence starts with the original image, taken from the highlighted area, which appears at a lower resolution, with less distinct features. As we progress through the sequence, each model attempts to improve the clarity and detail of the image. SRCNN, as one of the pioneering models in this field, enhances the image to a PSNR of 26.98 dB and an SSIM of 0.6578, VDSR [12] advances this further to a PSNR of 27.56 dB and an SSIM of 0.7367, while SRGAN achieves a PSNR of 28.00 dB and an SSIM of 0.7812. In Figure 4, Figure 6 and Figure 7, each of these results is visible as almost the same. Within this cohort, SRGAN exhibits the most commendable performance, achieving a PSNR of 28.45 dB and an SSIM of 0.8423, indicating its superior capability in enhancing image quality and structural fidelity. Following SRGAN, VDSR [12] attains a PSNR of 27.96 dB and an SSIM of 0.7865, showcasing its effective resolution enhancement features. Conversely, SRCNN, despite being an early innovator in this domain, records a PSNR of 27.14 dB and an SSIM of 0.6701. The baseline model, noted for its lightweight architecture, surprisingly yields a higher PSNR of 28.24 dB and an SSIM of 0.8103, demonstrating efficient performance despite its simplicity. However, the proposed model surpasses all these metrics, delivering the highest refinement with a PSNR of 28.87 dB and an SSIM of 0.8803.

Figure 5 presents a series of comparisons of our proposed model under noisy and low-contrast conditions. These examples highlight the model ability to mitigate noise and enhance subtle anatomical features, demonstrating its robustness and reliability for use in real-world clinical settings. The visual examples clearly show that our model effectively enhances image quality even in challenging situations, such as those with significant noise or low contrast, which are common in clinical practice. This further supports the suitability of our approach for medical applications where high fidelity is crucial.

### 4.5. Comparison of the Baseline Models

Table 2 offers a nuanced comparison of various super-resolution models, evaluating their performance across different scaling factors and showcasing their efficiency and effectiveness through parameters like runtime and the image quality metrics PSNR and SSIM. Dataset1 and Dataset2 refer to the Figshare and Kidney Stone datasets, respectively, which were used in our experiments. No additional datasets were introduced. We have explicitly stated this to ensure clarity and transparency in our manuscript. Furthermore, we performed several preprocessing modifications to the original datasets. These modifications included resizing the images to meet the input requirements of the proposed model, applying random augmentations such as rotations, flips, and brightness adjustments, and adding Gaussian noise to simulate realistic conditions. These preprocessing steps are detailed in the Data Preprocessing section to provide a comprehensive understanding of the dataset preparation.

At a 2× scale, models such as IMDN [35] and RFDN [36] demonstrate high-quality image enhancement, with IMDN [35] achieving notably high PSNR and SSIM scores. SRCNN, while less complex, with the fewest parameters, offers a fast processing time, making it efficient though slightly less effective in terms of the quality metrics (Figure 6). VDSR [12] and CARN [37], with their higher complexity, show improved image quality at the cost of an increased computational load. The proposed model distinguishes itself by achieving the best balance between runtime and high-quality results, outperforming other models in both datasets at this scale. When scaling up to 4×, all models generally experience a decrease in performance, indicative of the increased challenge associated with higher scaling factors.

However, the proposed model maintains robust performance, surpassing other models in image quality, which highlights its superior design for handling more significant upscaling challenges effectively. RFDN [36] and the baseline model also exhibit commendable performances, suggesting their potential utility in applications where a balance between speed and image quality is crucial. The diversity in the model performances across the two datasets underscores the importance of selecting a model based on specific application needs, considering factors such as the desired balance between image quality and computational efficiency. The proposed model, with its exceptional performance metrics, exemplifies the advancements in super-resolution technology, promising significant improvements in applications requiring detailed image resolution enhancements.

### 4.6. Comparison with SOTA Models

This section presents a comparison of the proposed RFLN with various SOTA models in medical imaging. The evaluation was performed on two distinct datasets, Dataset1 and Dataset2, using PSNR and SSIM as the primary metrics for assessing image quality. Additionally, the computational efficiency of each model was compared by measuring the runtime. The performance of the models at both 2× and 4× scaling factors is summarized in Table 2 and Table 3. The results highlight the superiority of the proposed RFLN model in preserving anatomical details, mitigating noise, and providing higher visual fidelity compared to previous methods. 

In Table 3, the proposed RFLN model achieves the highest PSNR and SSIM for both datasets. The model’s efficient residual learning and enhanced spatial attention ensure high-quality image reconstruction, while its runtime is the fastest among the compared models, demonstrating the balance between image quality and computational efficiency.

At the 4× scaling factor, as shown in Table 4, the RFLN model continues to outperform the other methods, particularly in PSNR, which measures the sharpness and fidelity of the super-resolved images. Figure 6 presents visual comparisons between the different models for a patch from a brain MRI scan from Dataset1. The images reconstructed by our model exhibit the best clarity and structural accuracy, with fewer artifacts compared to the other methods.

While all models show a decline in performance at higher scaling factors, the proposed RFLN model maintains its advantage in preserving image quality with fewer parameters and lower runtime, making it highly efficient for clinical deployment.

## 5. Discussion

The experimental results presented in this paper demonstrate the effectiveness of the enhanced RFLN for single image super-resolution in medical imaging. The proposed model consistently outperformed existing state-of-the-art models in both quantitative metrics, such as PSNR and SSIM, and qualitative visual assessments. This section provides a discussion of the key findings, their implications, and the limitations of the current approach. One of the significant findings of this study is the importance of enhancing feature extraction through the use of deeper residual blocks. By replacing the original residual local feature blocks with standard residual blocks and increasing the network depth, the model demonstrated a substantial improvement in its ability to extract and retain crucial features. This is particularly relevant for medical images, where capturing fine anatomical details is critical for accurate diagnosis. The ESA mechanism also played a vital role in improving the overall performance of the model by allowing it to focus more effectively on relevant regions of the input image, further boosting the quality of the super-resolved output. The integration of ESA proved to be highly beneficial in addressing one of the key challenges in medical imaging: distinguishing between essential anatomical features and noise. Medical images often contain regions with subtle differences that are critical for diagnosis. By incorporating ESA, the model was able to prioritize these regions, resulting in super-resolved images that not only had higher visual fidelity but also retained diagnostically important details. This is particularly useful in applications such as MRI and CT scans, where image clarity directly impacts clinical outcomes. The visual quality improvements provided by the proposed model have significant clinical implications, particularly in enhancing diagnostic accuracy. The enhanced resolution achieved by our method can lead to better detection of small pathologies, which are often challenging to identify in lower-resolution images. For instance, with CT scans, the model’s ability to preserve fine anatomical details can aid in the early detection of small lesions, which is critical for early-stage diagnosis and timely intervention. Similarly, in ultrasound imaging, a higher resolution can improve the visibility of subtle abnormalities, such as small kidney stones or early-stage tumors, leading to more accurate and confident diagnoses. By refining the visual quality of medical images, our model has the potential to support radiologists and other healthcare professionals in identifying pathologies that may otherwise go unnoticed. This enhancement not only improves diagnostic accuracy but also contributes to more effective treatment planning and patient outcomes. These clinical implications underscore the value of our approach beyond mere image quality improvement, highlighting its practical utility in supporting healthcare professionals with precise and reliable image enhancement tools.

Despite the promising results, there are several limitations to the current approach that warrant further investigation. First, while the proposed model shows significant improvement over existing methods, the computational complexity remains relatively high. The increased model depth and the incorporation of attention mechanisms add to the computational load, which could limit the model’s applicability in real-time clinical settings or in scenarios with limited computational resources. Future work could focus on optimizing the model to reduce inference time and computational requirements without compromising image quality. Another limitation is the reliance on specific datasets for training and evaluation. Although the proposed model performed well on the medical imaging datasets used in this study, the generalizability of the model to other types of medical images or imaging modalities needs to be further explored. Medical imaging data can vary significantly based on factors such as imaging equipment, acquisition protocols, and patient demographics. To ensure robustness and broad applicability, future research should include more diverse datasets and evaluate the model’s performance across them.

## 6. Conclusions

In this paper, we presented an enhanced version of the RFLN specifically designed for single image super-resolution in medical imaging. By incorporating deeper residual blocks, ESA mechanisms, and increasing the model depth, the proposed architecture demonstrated significant improvements in reconstructing high-resolution medical images from their low-resolution counterparts. Our experimental results showed that the enhanced RFLN outperformed existing state-of-the-art models in terms of both quantitative metrics, such as PSNR and SSIM, and qualitative visual assessments. The integration of ESA played a crucial role in focusing on relevant anatomical features while suppressing noise, leading to better diagnostic utility of the super-resolved images. These enhancements make the proposed model particularly well-suited for medical applications, where image quality directly impacts diagnostic accuracy and patient outcomes. However, the increased computational complexity and reliance on specific datasets present challenges that must be addressed in future work. Optimizing the model for real-time clinical deployment and expanding its applicability to diverse medical imaging modalities are essential next steps. By addressing these limitations, we believe that the proposed model can make a significant impact in the field of medical imaging, providing healthcare professionals with the tools needed for more precise and effective diagnoses. 

## Figures and Tables

**Figure 1 bioengineering-11-01179-f001:**
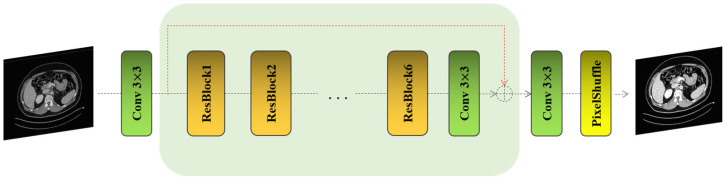
The architecture of the modified RFLN.

**Figure 2 bioengineering-11-01179-f002:**
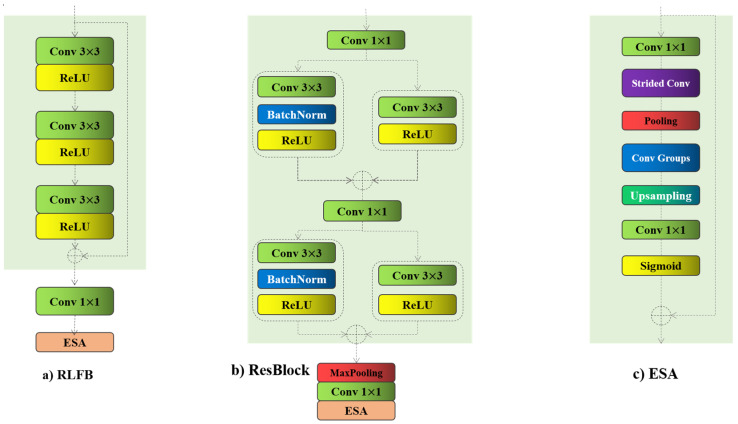
(**a**) RLFB: The residual local feature block; (**b**) ResBlock: Modified RLFB; (**c**) ESA: Enhanced Spatial Attention.

**Figure 3 bioengineering-11-01179-f003:**
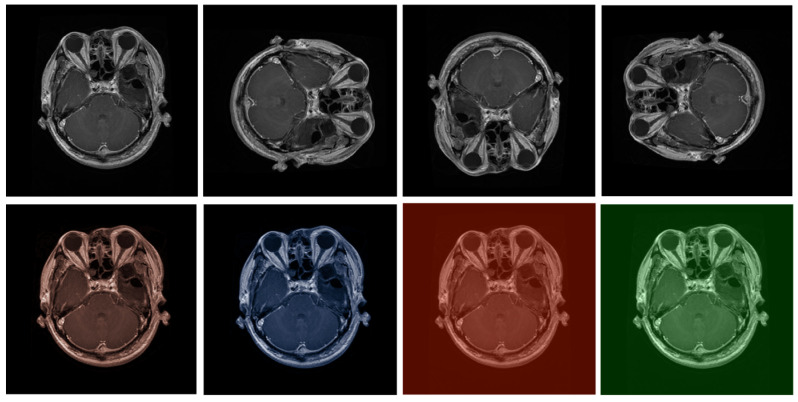
Data preprocessing.

**Figure 4 bioengineering-11-01179-f004:**
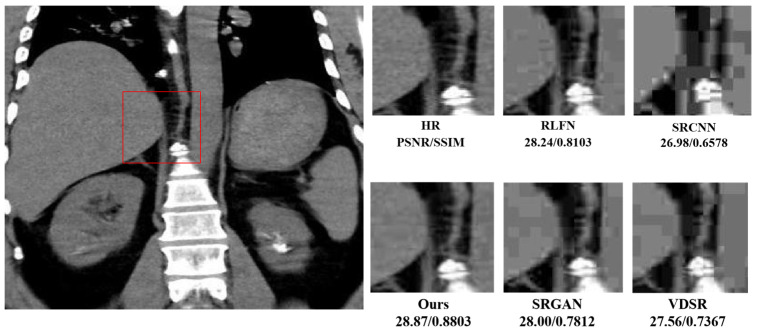
MRI images.

**Figure 5 bioengineering-11-01179-f005:**
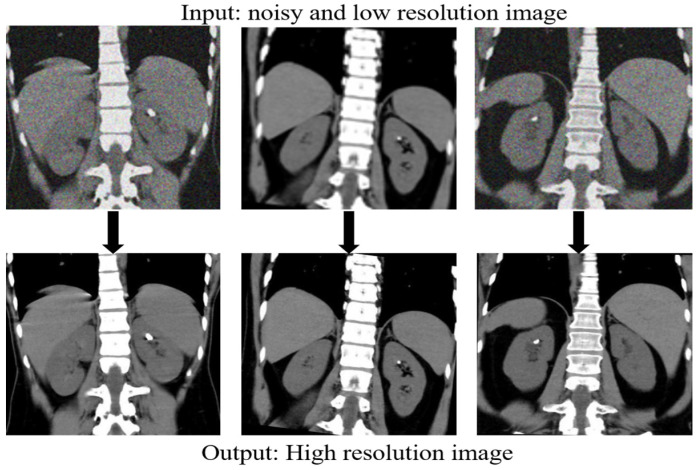
Presents a series of comparisons of our proposed model under noisy and low-contrast conditions.

**Figure 6 bioengineering-11-01179-f006:**
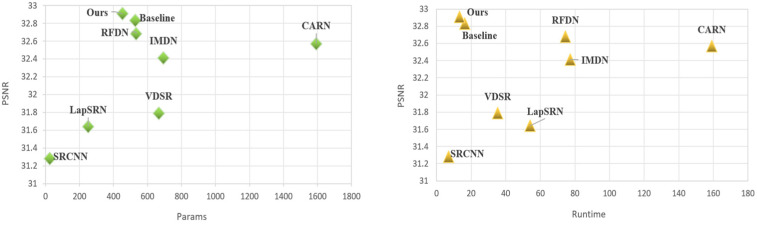
Illustration of the PSNR, Runtime, and Params for dataset1.

**Figure 7 bioengineering-11-01179-f007:**
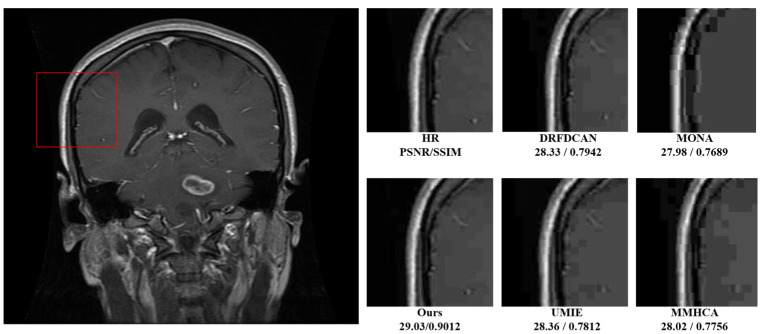
Visual comparison of the SOTA models.

**Table 1 bioengineering-11-01179-t001:** Characteristics of the Figshare and Kidney Stone datasets used in our experiments.

Dataset	Number of Images	Image Resolution	Modality	Description
Figshare	500	512 × 512	CT and MRI	A diverse set of medical images annotated for super-resolution tasks, covering multiple anatomical regions.
Kidney Stone	300	256 × 256	Ultrasound	A specialized dataset focusing on kidney stone images, annotated for improved clarity in super-resolution tasks.

**Table 2 bioengineering-11-01179-t002:** Results from the baseline models’ comparison.

Scale	Model	Params [K]	Runtime [ms]	Dataset1 PSNR/SSIM	Dataset2 PSNR/SSIM
	SRCNN [3]	24	6.92	31.28/0.9012	33.55/0.9312
	VDSR [12]	666	35.37	31.79/0.9056	33.78/0.9421
	IMDN [35]	694	77.34	32.41/0.9123	34.12/0.9512
2×	RFDN [36]	534	74.51	32.68/0.9154	34.45/0.9535
	CARN [37]	1592	159.10	32.57/0.9134	34.01/0.9486
	LapSRN [38]	251	53.98	31.64/0.9142	33.78/0.9356
	Baseline	527	16.41	32.83/0.9175	33.86/0.9369
	Ours	452	13.23	32.91/0.9188	34.07/0.9387
	SRCNN	57	1.90	27.78/0.7120	28.45/0.7276
	VDSR [12]	666	8.95	27.89/0.7165	28.61/0.7287
	IMDN [35]	715	20.56	27.95/0.7810	28.98/0.7301
	RFDN [36]	550	20.40	28.12/0.8023	29.23/0.7453
4×	CARN [37]	1592	39.96	27.86/0.7712	28.52/0.7282
	LapSRN [38]	502	66.81	27.15/0.6813	28.34/0.7145
	Baseline	543	16.41	28.34/0.8230	29.37/0.7478
	Ours	468	13.23	28.46/0.8256	29.48/0.7513

**Table 3 bioengineering-11-01179-t003:** Performance comparison at a 2× scaling factor.

Model	Params (K)	Runtime (ms)	Dataset1 PSNR/SSIM	Dataset2 PSNR/SSIM
MRI Super-Resolution [18]	1224	25.12	30.45/0.8921	32.78/0.9102
MONA [19]	1045	28.64	31.05/0.8998	33.22/0.9197
Fuzzy Neural Block [28]	880	22.51	30.12/0.8823	31.90/0.9051
MMHCA [29]	1342	55.78	31.56/0.9045	33.64/0.9228
UMIE [30]	1605	68.32	31.78/0.9107	33.85/0.9285
DRFDCAN [31]	953	26.91	32.14/0.9175	34.05/0.9322
Ours	452	13.23	32.91/0.9188	34.07/0.9387

**Table 4 bioengineering-11-01179-t004:** Performance comparison at a 4x scaling factor.

Model	Params (K)	Runtime (ms)	Dataset1 PSNR/SSIM	Dataset2 PSNR/SSIM
MRI Super-Resolution [18]	1224	25.12	27.12/0.7521	28.95/0.7779
MONA [19]	1045	28.64	27.98/0.7689	27.78/0.7901
Fuzzy Neural Block [28]	880	22.51	26.85/0.7456	28.62/0.7654
MMHCA [29]	1342	55.78	28.02/0.7756	27.86/0.7922
UMIE [30]	1605	68.32	28.36/0.7812	28.01/0.8027
DRFDCAN [31]	953	26.91	28.33/0.7942	29.22/0.8115
Ours	468	13.23	28.46/0.8256	29.48/0.8513

## Data Availability

The original contributions presented in the study are included in the article, further inquiries can be directed to the corresponding author/s.

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
