# Peer review of "Lightweight Super-Resolution Techniques in Medical Imaging: Bridging Quality and Computational Efficiency"

_bioengineering, 2024, doi:10.3390/bioengineering11121179_

Round 1

Reviewer 1 Report

Comments and Suggestions for Authors

The manuscript is devoted to the development of computer vision techniques in the field of medical image analysis. This problem fully corresponds to the aims and scope of the journal "Bioengineering".

There are the following comments:

1. At the end of Introduction, it is necessary to add a description of the further structure of the manuscript.

2. The volume of cited literature on the subject seems insufficient. It is recommended that it be increased to at least 40 items by reviewing more articles between 2023 and 2024.

3. In the text of the manuscript, before presenting the methodology, it is necessary to provide a formal statement of the problem that needs to be solved.

4. It is necessary to clearly justify the use of the ESA block and its architecture.

5. The characteristics of the Figshare and Kidney Stone datasets used for testing should be explicitly given in the text, for example, in the form of a table.

6. The title of Section 4.3 (line 277) must be left-aligned.

7. Figure 6 is clear, but the quality of the images should be significantly improved.

8. Are Dataset1 and Dataset2 the same as Figshare and Kidney Stone datasets, or are they subsets of them? If they are the same, it is better to state this explicitly. If modifications are made to the data, they should be clearly stated in the text of the manuscript.

Author Response

We sincerely thank the reviewers for their thorough and constructive feedback. Your comments have greatly contributed to improving the clarity, depth, and overall quality of our manuscript. We appreciate your insights and suggestions, which have enabled us to strengthen our work and provide a more comprehensive presentation of our research. Thank you for your time and effort in reviewing our paper. We present our response in file.

Reviewer 2 Report

Comments and Suggestions for Authors

The article presents a novel approach to Single Image Super-Resolution (SISR) in medical imaging by enhancing the Residual Feature Learning Network (RFLN) with deeper residual blocks and an Enhanced Spatial Attention (ESA) mechanism. The paper addresses a critical challenge in medical imaging -balancing image quality enhancement with computational efficiency - making it a valuable contribution to the field. However, some areas could benefit from further exploration and improvement to enhance the impact and generalizability of the work.

Strengths of the paper:

The introduction provides a comprehensive overview of SISR, its applications in medical imaging, and the limitations of existing techniques. Moreover, the motivation to conduct research in this area is well justified.

The paper explains the architectural modifications made to the baseline RFLN model, the enhanced spatial attention (ESA) mechanism, and the training methodology. Equations and diagrams are provided to clarify the structural changes and the underlying calculations. Thus, the research design is appropriate for the challenges of medical imaging and  the methods are adequately described.

The results are clearly presented. The quantitative evaluation, including PSNR and SSIM metrics, convincingly demonstrates the superiority of the proposed method over baseline and state-of-the-art models. The results highlight improvements in both image quality and computational efficiency. Also, the conclusions are well-supported by the results.

Weaknesses of the paper:

While the experiments utilize two specialized datasets (Figshare and Kidney Stone collections), the generalizability of the proposed model could be further validated by testing on a broader range of medical imaging modalities, such as CT, PET, or ultrasound scans. This would help demonstrate the model's adaptability to diverse clinical scenarios and equipment variability.

Although the manuscript includes visual comparisons, the analysis could be expanded to discuss the clinical implications of the visual quality improvements. For example, would the enhanced resolution lead to better detection of small pathologies or improve diagnostic accuracy in specific use cases?

The manuscript lacks a focus on challenging imaging scenarios, such as noisy or low-contrast inputs, which are common in clinical practice. Including visual examples of how the model performs in these scenarios would add practical relevance

Overall Assessment: The manuscript is a significant contribution to the field of medical imaging, addressing both the quality and computational efficiency of super-resolution models. It introduces meaningful architectural innovations and demonstrates their effectiveness through comprehensive experiments. However, testing on more diverse datasets and providing a deeper analysis of visual quality, particularly in clinically challenging scenarios, would further strengthen the work.

Author Response

We sincerely thank the reviewers for their thorough and constructive feedback. Your comments have greatly contributed to improving the clarity, depth, and overall quality of our manuscript. We appreciate your insights and suggestions, which have enabled us to strengthen our work and provide a more comprehensive presentation of our research. Thank you for your time and effort in reviewing our paper.We presents our response in file. 

Round 2

Reviewer 1 Report

Comments and Suggestions for Authors

All responses are clear. The manuscript can be accepted in its current form.